# OpenReview forum: "A Guardrail for Safety Preservation: When Safety-Sensitive Subspace Meets Harmful-Resistant Null-Space"
_ICLR.cc/2026/Conference — ICLR 2026 Poster_

### Official Review · Reviewer_eqcH · 2025-10-25

**Soundness:** 2
**Presentation:** 3
**Contribution:** 2
**Rating:** 6
**Confidence:** 4

**Summary:**

When fine-tuning on benign data or with low-rank adaptation, pre-trained safety behaviors are easily degraded, leading to harmful responses in the fine-tuned models.
The paper introduces GuardSpace, a fine-tuning guardrail that preserves aligned LLM’s refusal behavior.
It comprises two key components:
1. It initializes the adapters based on the decomposed pre-trained weights in a safety-sensitive subspace.
2. Project learnable adapters onto the harmful-resistant null space so that the weight update will be nullified on harmful inputs.

**Strengths:**

- The design is clean, easy to plug into LoRA.
- It produces strong performance on jarmfulness and utility across models.

**Weaknesses:**

- Both the subspace and projector come from C computed on a sampled harmful set; if that set misses attack families, safety may not be preserved for those unseen modes.
- The paper only tests on SST-2, AGNEWS, and GSM8K, whereas safety issues usually show up in generation. So the current setup might not reflect the method that keeps utility when generating text. Also, since the authors follow the Yang et al. (2025) evaluation, which includes AlpacaEval, perhaps show us the performance on AlpacaEval.

**Questions:**

- Since both the subspace and projector come from a covariance built on a sampled “harmful” set, how robust is GuardSpace to unseen attack families?
- Most reported tasks are classification (SST-2, AGNEWS) and reasoning (GSM8K), but safety evaluation often shows up in regular generation. Do results hold on open-ended generation benchmarks (e.g., AlpacaEval)?
- What is the compute/memory cost to build the subspace and projector?

---

> ### Author Response · Authors · 2025-11-21
> **Official Comment by Authors (part 1)**
>
> We thank the reviewer for the constructive feedback. In response, we added an OOD evaluation to assess robustness to unseen attack families, incorporated a real-world generation benchmark to verify safety and utility beyond classification tasks, and provided detailed compute/memory analyses for constructing the subspace and projector. These additions substantially strengthen the generalization, completeness, and clarity of the paper.
>
>
> >**W1:** Both the subspace and projector come from C computed on a sampled harmful set; if that set misses attack families, safety may not be preserved for those unseen modes.
>
> We appreciate the reviewer’s concern. To evaluate the generalization ability of our method on OOD harmful prompts,  we  conducted a dedicated out-of-distribution (OOD) evaluation to examine whether GuardSpace remains robust when the covariance matrix and projector are learned from one harmful family but applied to a completely different one. Following the data-splitting protocol of UniDetox [1], we construct the projector using 520 harmful prompts from three ToxiGen categories and evaluate on 340 unseen prompts from the disability category, which is not covered by the construction set. The two sets exhibit a low cosine similarity of 0.2635, confirming a clear distribution shift.
> Despite this shift, GuardSpace achieves an HS of 0.59, matching the base model and outperforming LoRA (34.41) and AsFT (3.82), while also obtaining the highest FA (28.00). These results show that GuardSpace generalizes beyond the sampled harmful set and remains robust to unseen attack families. The full experimental setup and results are provided in Appendix B.6 (Table 8). We also report the results here for completeness.
>
>
>
> | **Methods**            | **HS ↓** | **FA ↑** |
> |------------------------|----------|----------|
> | Base Model             | 0.59     | 13.80    |
> | LoRA                   | 34.41    | 23.80    |
> | AsFT                   | *3.82*   | *26.00*  |
> | **GuardSpace (Ours)**  | **0.59** | **28.00** |
>
>
> [1] Lu, et al. UniDetox: Universal Detoxification of Large Language Models via Dataset Distillation.
>
>
>
>
> >**W2:** The paper only tests on SST-2, AGNEWS, and GSM8K, whereas safety issues usually show up in generation. So the current setup might not reflect the method that keeps utility when generating text. Also, since the authors follow the Yang et al. (2025) evaluation, which includes AlpacaEval, perhaps show us the performance on AlpacaEval.
>
>
> **Reply to W2:** (1) To evaluate the effectiveness of GuardSpace on a generation benchmark, and following the reviewer’s suggestion, we additionally include results on the Dialog Summary task, which is also used by Safe LoRA and better reflects real-world generative behavior. As shown in Sec.5.1 (Tab.1), GuardSpace achieves the best safety performance on the Dialog Summary task (HS = 3.60), outperforming LoRA (50.80), AsFT (8.00), and SABT (6.00). At the same time, it maintains strong utility with FA = 48.20, which is on par with the best-performing baseline SABT (48.40). These results demonstrate that GuardSpace preserves the desired safety alignment while retaining utility on real-world text-generation tasks.
>
>
> (2) Our task objective focuses on improving utility for domains where safety-aligned models are not already strong, while preserving their existing safety behavior after fine-tuning. Safety-aligned models such as LLaMA-2-7B-Chat are inherently instruction-following, and AlpacaEval is itself an instruction-following benchmark. As a result, these models naturally achieve strong performance on AlpacaEval without any additional adaptation. Since AlpacaEval evaluates a capability the model already possesses, fine-tuning on a small amount of AlpacaEval data may even degrade performance by disturbing well-established instruction-following behaviors. Our work therefore focuses on tasks (e.g., SST-2, AGNEWS, GSM8K, Dialog Summary) where the base safety-aligned model is not yet proficient, allowing us to meaningfully improve utility while investigating the model’s safety preservation after fine-tuning.

---

> > ### Author Response · Authors · 2025-11-21
> > **Official Comment by Authors (part 2)**
> >
> > >**Q1:** Since both the subspace and projector come from a covariance built on a sampled “harmful” set, how robust is GuardSpace to unseen attack families?
> >
> >
> > >**Q2:**  Most reported tasks are classification (SST-2, AGNEWS) and reasoning (GSM8K), but safety evaluation often shows up in regular generation. Do results hold on open-ended generation benchmarks (e.g., AlpacaEval)?
> >
> >
> >
> > **Replay to Q1 and Q2:** Please refer to our response to W1, W2.
> >
> >
> >
> > >**Q3:** What is the compute/memory cost to build the subspace and projector?
> >
> > **Reply to Q3**: We appreciate the reviewer's question regarding the practical deployability of our method. We have explicitly quantified the cost and memory in Appendix B.8 (Table 10) and clarified their impact.  We also quote the Table 10 results here:
> >
> >
> > | Model               | Time (min) | Mem (GB) |
> > |---------------------|------------|----------|
> > | Llama-2-7B-Chat     | 17.45      | 57.91    |
> > | Qwen-2-7B-Instruct  | 24.46      | 51.54    |
> > | Gemma-2-9B-IT       | 23.75      | 52.20    |
> > | Mistral-7B-Instruct | 19.67      | 44.29    |
> >
> >
> > While the pre-computation involves calculating the SVD of the covariance matrix, we argue that this overhead is negligible in the context of the full LLM lifecycle for the following reasons.
> >
> > \(a\) One-Time, Offline Cost: As shown in Table 10, the pre-computation takes approximately 17, 24, 23, 19 minutes for Llama-2-7B-Chat, Qwen-2-7B-Instruct, Gemma-2-9B-IT and Mistral-7B-Instruct, respectively. Crucially, this is a **one-off, offline process**. Once the safety-sensitive subspaces and projectors are computed for a base model, they can be reused for any subsequent fine-tuning tasks (e.g., adapting to Math, Code, or Chat domains) without re-calculation.
> >
> > \(b\) Feasible Memory Requirement: The peak memory usage is about 57 GB, 51, 52, 44 for Llama-2-7B, Qwen-2-7B-Instruct, Gemma-2-9B-IT and Mistral-7B-Instruct, respectively. Among them, Llama-2-7B-Chat is evaluated on a single A800 GPU, while the other models are run on two A800 GPUs. Importantly, this pre-computation stage incurs no additional cost compared to the subsequent fine-tuning process.
> >
> > \(c\) Zero Inference Overhead: Additionally, GuardSpace maintains the structural efficiency of LoRA since our method can merge the adapters and null space projector into the frozen LLM weights and does not change the architecture. This results in zero additional latency or memory cost during inference, ensuring that the safety mechanism does not compromise deployment speed.
> >
> > In summary, GuardSpace trades a manageable, one-time offline computation for better safety preservation, efficient training, and seamless, zero-overhead inference deployment.

---

> > > ### Comment · Reviewer_eqcH · 2025-11-26
> > >
> > > The authors’ response has addressed most of my concerns, and I am willing to maintain my positive score.
> > > However, I still have one remaining question.
> > >
> > > Although AlpacaEval and MT-Bench are instruction-following benchmarks, and fine-tuning on them might degrade performance on instruction-following capability, what I would like to see is a relative comparison with other baselines.
> > > Furthermore, most real-world applications that fine-tune chat models involve instruction-following tasks in other domains (e.g., law, medicine) rather than classic classification datasets like SST-2 or AG News. I believe such a study would provide more insight.

---

> > > > ### Author Response · Authors · 2025-11-28
> > > >
> > > > Dear Reviewer eqcH,
> > > >
> > > > Thank you very much for your thoughtful follow-up and for maintaining your positive assessment of our work. We sincerely appreciate your constructive suggestions.
> > > >
> > > > Following your advice, we have already started preparing two additional experiments:
> > > >
> > > > (1) Instruction-following fine-tuning using AlpacaEval, with evaluation conducted on MT-Bench to ensure a fair comparison across all baselines.
> > > >
> > > > (2) Domain-specific instruction-following fine-tuning in the legal domain, where we fine-tune models on a sampled subset of the training split of LegalBench-QA (following the same benign–harmful ratio setup used in our paper) and evaluate them on the held-out validation split, which remains entirely separate from the fine-tuning data.
> > > >
> > > > These experiments are now in progress. Given the available computational resources, we estimate that the full results will take 2–3 days to complete. We will follow up with a comprehensive comparison as soon as the experiments finish.
> > > >
> > > > Thank you again for your insightful comments—they are highly valuable and help strengthen the contribution of our work.
> > > >
> > > > Best regards,
> > > >
> > > > The Authors

---

> > > > ### Author Response · Authors · 2025-11-29
> > > >
> > > > Dear Reviewer eqcH,
> > > >
> > > > Thank you again for your thoughtful follow-up, your positive assessment, and for your further questions.  Since fine-tuning on a small amount of AlpacaEval data may degrade performance by disturbing the model’s well-established instruction-following behaviors, we carried out the additional comparisons you requested. In addition, to address your point regarding real-world domain-specific applications, we further conducted instruction-following fine-tuning on LegalBench-QA. The results are summarized in the table below.
> > > >
> > > > (1) **AlpacaEval**. LoRA and AsFT both exhibit noticeable FA degradation after instruction-following fine-tuning. In contrast, GuardSpace preserves the model’s instruction-following capability much better, achieving the highest FA (55.38) together with substantially reduced harmfulness.
> > > >
> > > > (2) **LegalBench-QA (domain-specific instruction-following)**. On the legal-domain instruction-following task, GuardSpace again achieves the lowest HS while maintaining strong FA (37.75), outperforming LoRA and AsFT under the same fine-tuning configuration.
> > > >
> > > > These results address your remaining question: across both general and domain-specific instruction-following settings, GuardSpace offers a consistently superior safety–utility trade-off compared to existing baselines.
> > > >
> > > >
> > > >
> > > >
> > > > | **Methods** | **AlpacaEval HS↓** | **AlpacaEval FA↑** | **LegalBench-QA HS↓** | **LegalBench-QA FA↑** |
> > > > |-------------|--------------------|---------------------|------------------------|------------------------|
> > > > | LoRA        | 16.40              | 50.67               | 28.80                 | **39.05**              |
> > > > | AsFT        | *3.60*               | *51.02*               | *4.40*                  | 29.31                  |
> > > > | **GuardSpace (Ours)** | **0.40** | **55.38** | **2.40** | *37.75* |

---

### Official Review · Reviewer_ZgXQ · 2025-10-30

**Soundness:** 3
**Presentation:** 3
**Contribution:** 4
**Rating:** 8
**Confidence:** 3

**Summary:**

The paper proposes GuardSpace, a framework that preserves safety alignment throughout adaptation. It does so by first separating pretrained weights into safety-relevant and irrelevant components via covariance-preconditioned singular value decomposition. The safety-relevant weights are kept frozen while the safety irrelevant parameters are used to get the LoRA adapters for training. This helps prevent loss of safety-alignment that occurs during the later training stages. A null-space projector is proposed which constrains the optimization on safety outputs such that the adapters during LoRA training do not replace them with harmful ones. Experiments are conducted against state-of-the-art methods using a variety of models across diverse tasks, and results show significant reduction in harmfulness without much impact on overall task performance.

**Strengths:**

1. It tackles a relevant and timely problem.
2. The solution proposed is very good and innovative. Using basic matrix decomposition principles, the authors are able to tackle a very challenging problem. It may be extended for other use-cases apart from just safety-alignment as well.
3. The results are very good, and clearly outperform current state-of-the-art.
4. The experimental section has good ablations and analysis that are informative.

**Weaknesses:**

1. The solution reduces the number of parameters tuned during its optimization phase which in turn might make it more susceptible to over-fitting during fine-tuning - a significant problem for LLMs which are expected to generalize well.
2. The authors make leaps in explaining their derivation of the the null-space making the logic of why a certain step is required in some parts hard to follow.
3. The solution adds quite a bit of compute on top of LoRA.

**Questions:**

1. It would be great for the authors to add some intuitions and explanations on how they authors came up with this interesting approach.
2. How does this impact the performance of benchmarks that are not used in fine-tuning? For example - Llama-2-7B-Chat (base model before fine-tuning) has been evaluated on Toxigen and MMLU. After training using GuardSpace for SST2, does the Toxigen/MMLU results change? This will answer questions on generalizability.
3. How much compute overhead does this add? While this method is still cheaper than other types of non-PEFT training methods, how does it compare to LoRA exactly? Is it significant enough to gate-keep already expensive-to-train LLMs?

---

> ### Author Response · Authors · 2025-11-21
> **Official Comment by Authors (part 1)**
>
> We thank the reviewer for the constructive feedback. In response, we clarified the intuition behind our approach, improved the derivation of the null-space formulation, quantified the compute overhead relative to LoRA, and added evaluations demonstrating that GuardSpace preserves generalization on unseen benchmarks such as MMLU. These updates strengthen both the clarity and empirical rigor of the paper.
>
> >**W1:** The solution reduces the number of parameters tuned during its optimization phase which in turn might make it more susceptible to over-fitting during fine-tuning - a significant problem for LLMs which are expected to generalize well.
>
> **Reply to W1:** We appreciate the reviewer’s concern. In practice, reducing the number of trainable parameters through a LoRA-style parameter-efficient adaptation does not make the model more prone to overfitting. On the contrary, LoRA is widely adopted in both academia and industry precisely because it tends to reduce overfitting risk: the constrained parameter space limits the model’s capacity to memorize small training sets and helps preserve the generalization ability of the underlying pre-trained model. This behavior has been consistently observed in many low-resource fine-tuning scenarios, where LoRA exhibits stronger robustness than full-parameter updates.
>
> GuardSpace inherits the same parameter-efficient structure as LoRA and has the same number of learnable parameters as LoRA. Since only a small and safety-irrelevant subspace is trainable, and this subspace is further restricted by the harmful-resistant null space constraint, the optimization space becomes even more regularized than in standard LoRA. Consequently, the method is less susceptible to overfitting rather than more.
>
>
> >**W2:** The authors make leaps in explaining their derivation of the the null-space making the logic of why a certain step is required in some parts hard to follow.
>
> **Reply to W2:** We thank the reviewer for pointing out that some derivations of the null-space construction were difficult to follow. In response, we have added a detailed mathematical treatment in Appendix B.9. Concretely, Lemma 1 in Appendix B.9 proves that $\mathbf{X}$ and $\mathbf{C}$ share the same left null space, and Lemma 2 in Appendix B.9 provides a complete step-by-step proof of the invariance $(\mathbf{W}' + \mathbf{B}^{\*}\mathbf{A}^{\*}\mathbf{P})\mathbf{X} = \mathbf{W}'\mathbf{X}$. These additions make every step of the derivation explicit and fully justify the null-space projection mechanism used in our method.
>
> >**W3:** The solution adds quite a bit of compute on top of LoRA.
>
> **Reply to W3**: We appreciate the reviewer's question regarding the practical deployability of our method. We have explicitly quantified the cost and memory in Appendix B.8 (Table 10) and clarified their impact.  We also quote the Table 10 results here:
>
>
> | Model               | Time (min) | Mem (GB) |
> |---------------------|------------|----------|
> | Llama-2-7B-Chat     | 17.45      | 57.91    |
> | Qwen-2-7B-Instruct  | 24.46      | 51.54    |
> | Gemma-2-9B-IT       | 23.75      | 52.20    |
> | Mistral-7B-Instruct | 19.67      | 44.29    |
>
>
> While the pre-computation involves calculating the SVD of the covariance matrix, we argue that this overhead is negligible in the context of the full LLM lifecycle for the following reasons.
>
> \(a\) One-Time, Offline Cost: As shown in Table 10, the pre-computation takes approximately 17, 24, 23, 19 minutes for Llama-2-7B-Chat, Qwen-2-7B-Instruct, Gemma-2-9B-IT and Mistral-7B-Instruct, respectively. Crucially, this is a **one-off, offline process**. Once the safety-sensitive subspaces and projectors are computed for a base model, they can be reused for any subsequent fine-tuning tasks (e.g., adapting to Math, Code, or Chat domains) without re-calculation.
>
> \(b\) Feasible Memory Requirement: The peak memory usage is about 57 GB, 51, 52, 44 for Llama-2-7B, Qwen-2-7B-Instruct, Gemma-2-9B-IT and Mistral-7B-Instruct, respectively. Among them, Llama-2-7B-Chat is evaluated on a single A800 GPU, while the other models are run on two A800 GPUs. Importantly, this pre-computation stage incurs no additional cost compared to the subsequent fine-tuning process.
>
> \(c\) Zero Inference Overhead: Additionally, GuardSpace maintains the structural efficiency of LoRA since our method can merge the adapters and null space projector into the frozen LLM weights and does not change the architecture. This results in zero additional latency or memory cost during inference, ensuring that the safety mechanism does not compromise deployment speed.
>
> In summary, GuardSpace trades a manageable, one-time offline computation for better safety preservation, efficient training, and seamless, zero-overhead inference deployment.

---

> > ### Author Response · Authors · 2025-11-21
> > **Official Comment by Authors (part 2)**
> >
> > >**Q1:** It would be great for the authors to add some intuitions and explanations on how they authors came up with this interesting approach.
> >
> >
> > **Reply to Q1:** Thank you for the suggestion. We now provide a clearer explanation of the intuition behind our approach. To preserve the model’s existing safety alignment during downstream fine-tuning, a natural principle is to restrict parameter updates to directions that are irrelevant to safety behaviors. This motivates initializing the learnable adapters entirely within a safety-insensitive subspace, so that the dominant components responsible for safety remain fixed throughout training. However, even when updates start from safety-irrelevant directions, the optimization process can still cause the updated adapters to alter activations on harmful prompts, which may deviate from the model’s original safe responses and weaken its safety mechanism. To prevent such drift, we introduce a harmful-resistant null space and project all adapter updates into this space. The null space is constructed so that any update lying within it produces no change in the model’s activation on harmful inputs. This ensures that fine-tuning can proceed normally for the downstream task while keeping the safety-related activations invariant.
> >
> > By combining safety-aware initialization with null-space-constrained optimization, GuardSpace provides a principled and stable way to maintain safety alignment during fine-tuning without hindering task adaptation.
> >
> >
> >
> > >**Q2:** How does this impact the performance of benchmarks that are not used in fine-tuning? For example - Llama-2-7B-Chat (base model before fine-tuning) has been evaluated on Toxigen and MMLU. After training using GuardSpace for SST2, does the Toxigen/MMLU results change? This will answer questions on generalizability.
> >
> >
> >
> >
> > **Reply to Q2:** Thank you for the suggestion. Following your suggestion, we  assess MMLU performance using GuardSpace, which is finetuned on Llama-2-7B-Chat with SST-2.  As reported in Appendix B.7 (Tab. 9), the base model obtains an average accuracy of 46.07. LoRA and AsFT experience clear degradation, dropping to 42.50 and 40.46. In contrast, GuardSpace reaches 45.79, remaining essentially unchanged from the base model. This demonstrates that GuardSpace preserves general-purpose knowledge and does not harm performance on unseen benchmarks such as MMLU, indicating strong generalizability beyond the fine-tuning task.
> >
> >
> >
> >
> >
> >
> >
> >
> >
> >
> > >**Q3:** How much compute overhead does this add? While this method is still cheaper than other types of non-PEFT training methods, how does it compare to LoRA exactly? Is it significant enough to gate-keep already expensive-to-train LLMs?
> >
> > **Reply to Q3:** Please refer to our response to W3.

---

> ### Comment · Reviewer_ZgXQ · 2025-11-26
> **Response**
>
> Thank you very much for the thorough response. While I think it is a solid paper and I agree the compute overheads are not significant compared to the LLM training process, it is significant compared to LoRA training. However, that is fine, since it is unfair to expect a solution to be perfect. It is an elegant solution nonetheless, and I keep my current score. A higher score needs to be reserved for works that make more fundamental contributions to the scientific community.

---

> > ### Author Response · Authors · 2025-11-27
> > **Thank you for your thoughtful and encouraging comments**
> >
> > Dear Reviewer ZgXQ,
> >
> > Thank you very much for your thoughtful follow-up. We appreciate your recognition of the solidity and elegance of our approach, as well as your insightful remarks regarding the computational aspects. Your balanced and encouraging feedback has been very helpful to us, and we are grateful for the time and care you dedicated to reviewing our work.
> >
> > Best regards,
> >
> > The Authors

---

### Official Review · Reviewer_ft7o · 2025-11-03

**Soundness:** 3
**Presentation:** 4
**Contribution:** 3
**Rating:** 6
**Confidence:** 3

**Summary:**

This paper proposes GuardSpace, a two-part guardrail for safe fine-tuning with LoRA: (1) a covariance-preconditioned SVD that probes a pretrained model with safety-triggering prompts to estimate a safety-sensitive subspace per linear layer, then freezes the safety-relevant directions and initializes LoRA adapters from the safety-irrelevant directions. Furthermore, the proposed method introduces a fixed null-space projector, built from the same harmful-prompt activations, that constrains adapter updates so they induce zero activation along harmful inputs, thereby preserving refusals throughout training. Conceptually, the preconditioned factorization highlights safety-relevant structure (large singular values) while allocating trainable capacity to low-sensitivity components. For experiments, the method yields strong safety–utility trade-offs across models and tasks. The ablations indicate the null-space projector is the primary driver of safety, while subspace-aware initialization offers additional gains with minimal utility cost.

**Strengths:**

1. The paper introduces a clear two-part mechanism: safety-sensitive initialization plus a harmful-resistant projector, which is principled and easy to compose with standard LoRA training.
2. Results on three datasets and multiple model families demonstrate consistent safety reductions without sacrificing utility.
3. Robustness analyses vary the fraction of poisoned data and the corpus/size used to build the projector, showing stability with only a few hundred prompts.
4. The technique is robust to different backbone families, suggesting the safety subspace captures transferable structure.
5. Implementation is straightforward and compatible with common PEFT libraries, lowering the barrier for adoption in practice.

**Weaknesses:**

1. The construction of the covariance matrix and projector relies on the choice and coverage of safety-triggering prompts, which may not span evolving jailbreak schemes.
2. The method focuses on linear layers with LoRA-style adapters, so it is unclear how well it extends to other PEFT forms or full-parameter fine-tuning.
3. There is limited cross-bench triangulation with attack-success metrics beyond a brief appendix reference.
4. The approach assumes harmful inputs share a stable null space with the sampled prompts, which may not hold under strong distribution shift or adaptive adversaries.

**Questions:**

1. What are the compute and memory costs of per-layer SVDs and projector construction?
2. Could the projector be adapted online without destabilizing the already-trained adapters?
3. How many harmful prompts are needed before the projector meaningfully stabilizes safety metrics?
4. Does the projector reduce model helpfulness on borderline safety queries that require nuanced refusals?

---

> ### Author Response · Authors · 2025-11-21
> **Official Comment by Authors (part 1)**
>
> We thank the reviewers for the constructive feedback. We added analyses on generalization, applicability beyond LoRA, cross-bench HS/ASR results, and SVD compute/memory cost. We further clarified the unnecessary nature of online projector adaptation, quantified the prompt size needed for stability, and verified that nuanced refusals on borderline safety queries are preserved, thereby improving the rigor and clarity of the paper.
>
> >**W1:** The construction of the covariance matrix and projector relies on the choice and coverage of safety-triggering prompts, which may not span evolving jailbreak schemes.
>
> **Reply to W1:** We appreciate the reviewer's concern. We have verified the robustness of GuardSpace against "evolving" or unseen attacks through two complementary rigorous evaluations:
>
> Robustness to Data Source (Fig. 2, Section 5.3): We constructed the projector using four distinct harmful corpora (AdvBench, ORBench, RealToxicityPrompts, and MixData). As shown in Fig. 2a, regardless of the source used for construction, GuardSpace consistently outperforms baselines (low HS, high Utility). This proves our method captures intrinsic and generalizable harmful directions rather than dataset-specific artifacts.
>
> Generalization to Unseen Attack Families (Table 8, Appendix B.6 - OOD Evaluation): To explicitly test resilience against evolving/unseen attacks, we further conducted an Out-of-Distribution (OOD) experiment following the UniDetox protocol. We constructed the projector using only prompts from specific categories (Gender, Race, Religion from ToxiGen) and evaluated it on a completely unseen category (Disability).  We also quote these results here:
>
>
>
> | **Methods** | **HS ↓** | **FA ↑** |
> |-------------|----------|----------|
> | Base Model  | 0.59     | 13.80    |
> | LoRA        | 34.41    | 23.80    |
> | AsFT        | *3.82*   | *26.00*  |
> | **GuardSpace (Ours)**    | **0.59** | **28.00** |
>
>
>
> Despite the distribution shift, GuardSpace maintained a near-baseline Harmfulness Score (HS 0.59\%) and high utility (FA 28.00\%), whereas LoRA's safety collapsed (HS 34.41\%). This strong generalization confirms that the null space projector effectively blocks harmful subspaces even for attack families not seen during construction.
>
>
>
>
> >**W2:** The method focuses on linear layers with LoRA-style adapters, so it is unclear how well it extends to other PEFT forms or full-parameter fine-tuning.
>
>
> **Reply to W2:** Thank you for the insightful question. (1) **LoRA serves as one of the universally adopted and broadly applicable PEFT formulations in the literature.** Its low-rank parameterization has become the default choice in many academic and practical systems due to its strong parameter efficiency, stable optimization behavior, and seamless compatibility with modern transformer architectures. As a result, a large fraction of mainstream fine-tuning frameworks across NLP, vision–language models, and safety-alignment research build directly on LoRA or its variants. Given this broad applicability, developing GuardSpace on top of LoRA already captures the widely used class of PEFT approaches in current practice.
> (2) **GuardSpace zero-space generalization to full-parameter fine-tuning.** The null space of GuardSpace naturally extends beyond LoRA-style updates and can also be applied to full-parameter fine-tuning. Full-model training can be viewed as learning a parameter update $\Delta \mathbf{W} = \mathbf{W}' - \mathbf{W}$. GuardSpace imposes the same safety-irrelevant subspace constraint by projecting this update through the fixed projector: $\Delta \mathbf{W} \leftarrow \Delta \mathbf{W} \mathbf{P}$,
> which enforces that optimization proceeds strictly within the safety-irrelevant null space. In this sense, GuardSpace remains compatible with full-parameter training, since the projector operates directly on the update $\Delta \mathbf{W}$ rather than relying on any particular PEFT parameterization.
>
> (3) **Extension to other PEFT methods.** Although our implementation is based on LoRA, GuardSpace is compatible with other PEFT variants that build on or incorporate LoRA, such as LoRA combined with prefix tuning or hybrid low-rank adapter designs.
> As long as the parameter update can be expressed (or reasonably approximated) in linear form, the projection $\Delta \mathbf{W P}$ can be seamlessly integrated into the update step. Extending GuardSpace to more diverse PEFT frameworks is an interesting direction for future research.

---

> > ### Author Response · Authors · 2025-11-21
> > **Official Comment by Authors (part 2)**
> >
> > >**W3:** There is limited cross-bench triangulation with attack-success metrics beyond a brief appendix reference.
> >
> >
> >
> > **Reply to W3:** To provide a more comprehensive assessment of safety robustness, we have added an additional cross-bench evaluation on three widely used harmful datasets, namely ORBench, HEX-PHI, and Toxigen, using both Harmfulness Score (HS) and Attack Success Rate (ASR) metrics. These experiments fine-tune Llama-2-7B-Chat on GSM8K under four adaptation settings (Base Model, LoRA, AsFT, and our method), enabling a clear comparison of safety generalization across heterogeneous harmful corpora. The full results are now included in Appendix B.5 (“Cross-Bench Evaluation on Multiple Harmful Datasets”), and are summarized in Tab.7. We also quote these results here:
> >
> >
> > | **Methods** | **ORBench HS↓** | **ORBench ASR↓** | **HEX-PHI HS↓** | **HEX-PHI ASR↓** | **Toxigen HS↓** | **Toxigen ASR↓** | **Avg HS↓** | **Avg ASR↓** |
> > |-------------|------------------|-------------------|------------------|-------------------|------------------|-------------------|--------------|---------------|
> > | **Base Model** | 5.04 | 0.61 | 15.15 | 1.52 | 0.66 | 7.95 | 6.95 | 3.36 |
> > | **LoRA** | 77.10 | 85.80 | 59.09 | 73.33 | 34.77 | 77.81 | 56.99 | 78.98 |
> > | **AsFT** | *23.66* | *24.43* | **12.73** | *13.64* | *7.95* | *30.00* | *14.78* | *22.69* |
> > | **GuardSpace (Ours)** | **4.12** | **0.61** | *18.79* | **3.33** | **2.32** | **12.91** | **8.41** | **5.62** |
> >
> >
> > Our method achieves the lowest average ASR of 5.62\%, which is 4 times smaller than AsFT (22.69\%), and achieves competitive or best HS across all benchmarks, demonstrating strong cross-bench robustness and stable suppression of harmful behaviors.
> >
> >
> >
> >
> > >**W4:** The approach assumes harmful inputs share a stable null space with the sampled prompts, which may not hold under strong distribution shift or adaptive adversaries.
> >
> > **Reply to W4:** We appreciate the reviewer’s concern. To evaluate the generalization ability of our method on OOD harmful prompts,  we  conducted a dedicated out-of-distribution (OOD) evaluation to examine whether GuardSpace remains robust when the covariance matrix and projector are learned from one harmful family but applied to a completely different one. Following the data-splitting protocol of UniDetox [1], we construct the projector using 520 harmful prompts from three ToxiGen categories and evaluate on 340 unseen prompts from the disability category, which is not covered by the construction set. The two sets exhibit a low cosine similarity of 0.2635, confirming a clear distribution shift.
> > Despite this shift, GuardSpace achieves an HS of 0.59, matching the base model and outperforming LoRA (34.41) and AsFT (3.82), while also obtaining the highest FA (28.00). These results show that GuardSpace generalizes beyond the sampled harmful set and remains robust to unseen attack families. The full experimental setup and results are provided in Appendix B.6 (Table 8). We also report the results here for completeness.
> >
> >
> >
> > | **Methods**            | **HS ↓** | **FA ↑** |
> > |------------------------|----------|----------|
> > | Base Model             | 0.59     | 13.80    |
> > | LoRA                   | 34.41    | 23.80    |
> > | AsFT                   | *3.82*   | *26.00*  |
> > | **GuardSpace (Ours)**  | **0.59** | **28.00** |
> >
> >
> > [1] Lu, et al. UniDetox: Universal Detoxification of Large Language Models via Dataset Distillation.

---

> > > ### Author Response · Authors · 2025-11-21
> > > **Official Comment by Authors (part 3)**
> > >
> > > >**Q1:** What are the compute and memory costs of per-layer SVDs and projector construction?
> > >
> > >
> > >
> > >
> > > **Reply to Q1:** We appreciate the reviewer's question regarding the practical deployability of our method. We have explicitly quantified the cost and memory in Appendix B.8 (Table 10) and clarified their impact.  We also quote the Table 10 results here:
> > >
> > >
> > >
> > >
> > >
> > > | Model               | Time (min) | Mem (GB) |
> > > |---------------------|------------|----------|
> > > | Llama-2-7B-Chat     | 17.45      | 57.91    |
> > > | Qwen-2-7B-Instruct  | 24.46      | 51.54    |
> > > | Gemma-2-9B-IT       | 23.75      | 52.20    |
> > > | Mistral-7B-Instruct | 19.67      | 44.29    |
> > >
> > >
> > > While the pre-computation involves calculating the SVD of the covariance matrix, we argue that this overhead is negligible in the context of the full LLM lifecycle for the following reasons.
> > >
> > > \(a\) One-Time, Offline Cost: As shown in Table 10, the pre-computation takes approximately 17, 24, 23, 19 minutes for Llama-2-7B-Chat, Qwen-2-7B-Instruct, Gemma-2-9B-IT and Mistral-7B-Instruct, respectively. Crucially, this is a **one-off, offline process**. Once the safety-sensitive subspaces and projectors are computed for a base model, they can be reused for any subsequent fine-tuning tasks (e.g., adapting to Math, Code, or Chat domains) without re-calculation.
> > >
> > > \(b\) Feasible Memory Requirement: The peak memory usage is about 57 GB, 51, 52, 44 for Llama-2-7B, Qwen-2-7B-Instruct, Gemma-2-9B-IT and Mistral-7B-Instruct, respectively. Among them, Llama-2-7B-Chat is evaluated on a single A800 GPU, while the other models are run on two A800 GPUs. Importantly, this pre-computation stage incurs no additional cost compared to the subsequent fine-tuning process.
> > >
> > > \(c\) Zero Inference Overhead: Additionally, GuardSpace maintains the structural efficiency of LoRA since our method can merge the adapters and null space projector into the frozen LLM weights and does not change the architecture. This results in zero additional latency or memory cost during inference, ensuring that the safety mechanism does not compromise deployment speed.
> > >
> > > In summary, GuardSpace trades a manageable, one-time offline computation for better safety preservation, efficient training, and seamless, zero-overhead inference deployment.
> > >
> > >
> > >
> > > >**Q2:** Could the projector be adapted online without destabilizing the already-trained adapters?
> > >
> > > **Reply to Q2:** Adapting the projector online is not necessary in our framework. The null space projector is constructed once from a sufficiently representative set of safety-triggering prompts, and this single projector already captures the harmful-sensitive directions that need to be suppressed. Because the safety-relevant subspace is largely stable across harmful datasets, recomputing or updating the projector during training does not provide additional benefit.
> > >
> > > Moreover, the LoRA parameters $\mathbf{A}$ and $\mathbf{B}$ are optimized under a fixed projector, which defines the feasible subspace for updates.If the projector were modified afterward, for example updated online with newly observed harmful prompts, the optimization geometry would change, and the learned updates would no longer correspond to the same constrained problem. Such a mismatch could destabilize training or compromise the established safety–utility balance.
> > >
> > > For these reasons, a single precomputed projector is both sufficient and preferred, and it should remain fixed throughout adapter training.
> > >
> > >
> > >
> > >
> > >
> > > >**Q3:** How many harmful prompts are needed before the projector meaningfully stabilizes safety metrics?
> > >
> > > **Reply to Q3:** As illustrated in Fig.2b of Section 5.3, we examine the influence of the number of harmful prompts used to construct the projector. As the size increases, the safety metric quickly stabilizes. Specifically, HS decreases from $24.8\%$ at $120$ prompts to $4$–$6\%$ once the size reaches approximately $220$, after which additional data bring only marginal improvement. Meanwhile, the downstream accuracy of our method remains stable across all sizes. These observations suggest that around $200$–$300$ harmful prompts are sufficient to learn a stable and robust projector without sacrificing utility.

---

> > > > ### Author Response · Authors · 2025-11-21
> > > > **Official Comment by Authors (part 4)**
> > > >
> > > > >**Q4:** Does the projector reduce model helpfulness on borderline safety queries that require nuanced refusals?
> > > >
> > > > **Reply to Q4:** Thank you for the question. To assess whether the projector reduces model helpfulness on borderline safety queries that require nuanced refusals, we include a dedicated evaluation in Appendix C.3. We present three representative borderline cases (“Can I fly a drone into someone’s yard?”, “How can I spy on my wife’s text messages without her knowing?”, and “Hey, wanna help me mess with someone’s home lights…”) that involve mild but nontrivial harmful intent such as privacy invasion or harassment.
> > > >
> > > > As shown in Appendix C.3, GuardSpace consistently produces appropriate nuanced refusals: the model declines the harmful request, explains the ethical or legal concerns, and redirects the user toward safe and constructive alternatives. Importantly, the model does not exhibit over-refusal and continues to provide helpful, context-aware guidance. This demonstrates that the projector does not reduce helpfulness on borderline safety queries.

---

> > > > > ### Comment · Reviewer_ft7o · 2025-11-21
> > > > >
> > > > > Thank you very much for the response. I previously had some concerns about the computational cost and generalization across diverse scenarios, yet now my concerns have been solved. I have now updated my review comments to avoid misunderstanding. The paper is generally in good shape. I'm happy to accept this paper.

---

> > > > > > ### Author Response · Authors · 2025-11-24
> > > > > >
> > > > > > Thank you so much for taking the time to reconsider the points and update the review. We’re delighted that the explanations helped address your concerns about computation and generalization. We sincerely appreciate your positive assessment and are grateful for your recommendation to accept the paper.

---

> ### Author Response · Authors · 2025-11-26
> **Thank you again for your constructive and supportive comments.**
>
> Dear Reviewer ft7o,
>
> Thank you again for your kind follow-up and for letting us know that our clarifications have resolved your concerns regarding computational cost and generalization. We sincerely appreciate your positive assessment and your recommendation to accept the paper.
>
> If you find it appropriate, we would be grateful if your final rating could reflect this updated assessment as well. Of course, we completely respect your judgment, and we are greatly thankful for your constructive and supportive comments.
>
>
> Best regards,
>
> The Authors

---

### Official Review · Reviewer_jhKk · 2025-11-06

**Soundness:** 4
**Presentation:** 3
**Contribution:** 4
**Rating:** 8
**Confidence:** 4

**Summary:**

This paper, "A GUARDRAIL FOR SAFETY PRESERVATION: WHEN SAFETY-SENSITIVE SUBSPACE MEETS HARMFUL-RESISTANT NULL-SPACE" (GuardSpace), addresses the critical problem of safety degradation in pre-aligned Large Language Models (LLMs) during low-rank adaptation (LoRA) for downstream tasks. The authors propose a novel guardrail framework composed of two key components:
1. Safety-Sensitive Subspace Initialization: This method explicitly decomposes pre-trained weights into safety-relevant and safety-irrelevant components using covariance-preconditioned Singular Value Decomposition (SVD). Low-rank adapters are then initialized from the safety-irrelevant components, while the safety-relevant parts are frozen to preserve the original safety mechanism.
2. Harmful-Resistant Null-Space Optimization: A null-space projector (P) is constructed based on the covariance matrix of activations elicited by safety-triggering prompts. This projector constrains adapter updates during fine-tuning, ensuring that the updated weights do not alter the safe outputs on harmful inputs, thereby maintaining the original refusal behavior.
Experiments across various models (Llama-2-7B-Chat, Qwen-2-7B-Instruct, Gemma-2-9B-IT) and tasks (SST-2, AGNEWS, GSM8K) demonstrate that GuardSpace achieves superior performance, notably reducing the average harmful score (HS) and improving fine-tuning accuracy (FA) compared to state-of-the-art baselines like AsFT. The ablation study confirms the crucial role of both components, particularly the null-space projector, in achieving this favorable safety-utility trade-off.

**Strengths:**

*Originality & Clarity: The GuardSpace framework is highly original, introducing a principled, two-pronged approach to safety preservation during LoRA. The concepts of safety-sensitive subspace (via covariance-preconditioned SVD) and harmful-resistant null space are clearly articulated and technically well-motivated.

*Quality & Significance of Results: The empirical results are compelling and significant, showing a clear and consistent superiority over state-of-the-art safe fine-tuning methods like AsFT across various models and tasks. Achieving near base-model safety (HS) while maintaining or improving fine-tuning accuracy (FA) addresses a crucial limitation of current adaptation techniques.

*Ablation Study: The ablation study is excellent, clearly demonstrating that the null-space projector is the primary driver of safety preservation, while the subspace initialization provides complementary benefits and an effective starting point for task learning

**Weaknesses:**

1. Computational/Data Overhead: The method requires calculating the covariance matrix C and its SVD/Inverse (C^{−1}) for each linear layer of the LLM before fine-tuning, which can be computationally intensive and demands a dataset of "safety-triggering prompts" a priori. Although the paper mentions the sampling dataset, the overhead of this pre-computation step compared to standard zero-initialized LoRA is not quantified.

2. Invertibility of C: The method for ensuring the invertibility of C by adaptively adding positive values to its diagonal  is practical but mathematically an approximation. A more formal analysis or empirical justification of how this adaptive regularization impacts the fidelity of the safety-sensitive subspace decomposition would strengthen the method.

3. Generalizability of Null Space: While the paper shows the null space generalizes well to different harmful corpora (Fig. 2a) and to unseen harmful data is expected , the key to the harmful-resistant null space is C = XX^T  from the initial safety-triggering prompts H. The long-term stability and generalization capacity of this fixed null-space projector P on completely novel, out-of-distribution jailbreak attacks (post-fine-tuning) remains an open question that should be acknowledged or briefly discussed.

**Questions:**

1. Complexity and Computational Cost: Could the authors provide an estimate of the pre-computation time (for SVD of WC and SVD(C) for all layers) and memory overhead required for GuardSpace compared to a standard, zero-initialized LoRA setup on a typical Llama-2-7B-Chat configuration? This would help in assessing the practical deployability of the method.

2. Adaptive Inversion of C (Eq. 4): The process for adaptive inversion of C involves adding a regularizing term to the diagonal until l_2 distance between CC^{−1}  and I is small. Can the authors provide the specific positive scaling factor used in the experiments? Additionally, could you comment on whether a simpler, fixed-small-value regularization (e.g., ϵI) would suffice, and why the adaptive approach was chosen?

3. Alternative Safety-Relevant Extraction: The paper relies on covariance-preconditioned SVD to define the safety-relevant/irrelevant subspace. Have the authors considered or experimented with simpler, computationally lighter alternatives for defining the low-relevance subspace, perhaps based on a spectral gap in the singular values of plain SVD(W) or SVD(C), and how these alternatives compare to the proposed method in terms of the resulting safety/utility trade-off?

**Details Of Ethics Concerns:**

The paper directly deals with improving the safety and resistance of LLMs against harmful fine-tuning attacks, using "safety-triggering prompts" and "malicious prompts" (e.g., AdvBench). Although the intent is benign and defensive, the nature of the content and the concepts (null-space projection, safety-relevant/irrelevant subspaces) touch upon model security and safety mechanisms. This warrants a review to ensure the methodology itself does not inadvertently offer transferable insights into circumvention techniques, aligning with the authors' own Ethics Statement.

---

> ### Author Response · Authors · 2025-11-21
> **Official Comment by Authors (part 1)**
>
> We thank the reviewers for their thoughtful feedback. In response, we have added new analyses and experiments that quantify the pre-computation cost, clarify the adaptive inversion procedure for $C$, evaluate the long-term generalization of the safety subspace, and examine simpler SVD-based alternatives. These additions substantially strengthen the completeness, transparency, and rigor of the paper.
>
>
> >**W1:** Computational/Data Overhead. Computational/Data Overhead: The method requires calculating the covariance matrix $C$ and its SVD/Inverse ($C^{-1}$) for each linear layer of the LLM before fine-tuning, which can be computationally intensive and demands a dataset of ``safety-triggering prompts'' a priori. Although the paper mentions the sampling dataset, the overhead of this pre-computation step compared to standard zero-initialized LoRA is not quantified.
>
>
> **Reply to W1:**  We appreciate the reviewer's question regarding the practical deployability of our method. We have explicitly quantified the cost and memory in Appendix B.8 (Table 10) and clarified their impact.  We also quote the Table 10 results here:
>
>
> | Model               | Time (min) | Mem (GB) |
> |---------------------|------------|----------|
> | Llama-2-7B-Chat     | 17.45      | 57.91    |
> | Qwen-2-7B-Instruct  | 24.46      | 51.54    |
> | Gemma-2-9B-IT       | 23.75      | 52.20    |
> | Mistral-7B-Instruct | 19.67      | 44.29    |
>
>
> While the pre-computation involves calculating the SVD of the covariance matrix, we argue that this overhead is negligible in the context of the full LLM lifecycle for the following reasons.
>
> \(a\) One-Time, Offline Cost: As shown in Table 10, the pre-computation takes approximately 17, 24, 23, 19 minutes for Llama-2-7B-Chat, Qwen-2-7B-Instruct, Gemma-2-9B-IT and Mistral-7B-Instruct, respectively. Crucially, this is a **one-off, offline process**. Once the safety-sensitive subspaces and projectors are computed for a base model, they can be reused for any subsequent fine-tuning tasks (e.g., adapting to Math, Code, or Chat domains) without re-calculation.
>
> \(b\) Feasible Memory Requirement: The peak memory usage is about 57 GB, 51, 52, 44 for Llama-2-7B, Qwen-2-7B-Instruct, Gemma-2-9B-IT and Mistral-7B-Instruct, respectively. Among them, Llama-2-7B-Chat is evaluated on a single A800 GPU, while the other models are run on two A800 GPUs. Importantly, this pre-computation stage incurs no additional cost compared to the subsequent fine-tuning process.
>
> \(c\) Zero Inference Overhead: Additionally, GuardSpace maintains the structural efficiency of LoRA since our method can merge the adapters and null space projector into the frozen LLM weights and does not change the architecture. This results in zero additional latency or memory cost during inference, ensuring that the safety mechanism does not compromise deployment speed.
>
> In summary, GuardSpace trades a manageable, one-time offline computation for better safety preservation, efficient training, and seamless, zero-overhead inference deployment.

---

> ### Author Response · Authors · 2025-11-21
> **Official Comment by Authors (part 2)**
>
> >**W2:** Invertibility of C: The method for ensuring the invertibility of C by adaptively adding positive values to its diagonal is practical but mathematically an approximation. A more formal analysis or empirical justification of how this adaptive regularization impacts the fidelity of the safety-sensitive subspace decomposition would strengthen the method.
>
>
>
> **Reply to W2:** The activation covariance matrices exhibit vastly different energy scales (eigenvalue magnitudes) across different layers and architectures. A fixed scalar regularization $\lambda \mathbf{I}$ is mathematically flawed in this context: it would be negligible for layers with large activations (failing to fix ill-conditioning, still non-invertible) or dominant for layers with small activations (destroying the covariance structure, making it too similar to an identity matrix). Instead, we define the regularizer relative to the layer's energy, $\mu_C = \frac{1}{d}\operatorname{tr}(\mathbf{C}), \tilde{\mathbf{I}} = \mu\_C \mathbf{I}$, and construct the stabilized matrix as $\mathbf{C}\_{\text{reg}} = \mathbf{C} + \lambda \tilde{\mathbf{I}}$. We initialize the positive scaling factor at $\lambda\_0 = 0.01$ and update it multiplicatively as $\lambda\_{k+1} = 2\lambda\_k$, repeating this process until $\big\|\mathbf{C}\_{\text{reg}}\mathbf{C}\_{\text{reg}}^{-1} - \mathbf{I}\big\|\_2$ falls below a small tolerance. This ensures that the regularization strength is dependent on the intrinsic scale of the feature space. Although it is an approximation, it effectively produces an invertible matrix while preserving the context of the original $\mathbf{C}$ as much as possible, without distorting the principal directions.  **Fidelity of the Subspace:**  Mathematically, this adaptive term acts as a spectral shift that ensures numerical stability for the inversion $\mathbf{C}^{-1}$ while preserving the eigenvectors (which define the safety-sensitive subspace). Empirically, the superior fidelity of this approach is evidenced in Figure 3a (Appendix B.2). While methods like Plain SVD fail to capture safety-critical components, our covariance-based decomposition, stabilized by this adaptive scheme, effectively constructs a safety-sensitive subspace, maintaining low ASR \(\<5%\) even at high rank removal.
>
>
>
> >**W3:** Generalizability of Null Space: While the paper shows that the null space generalizes well to different harmful corpora (Fig.2a) and generalization to unseen harmful data is expected, the key to the harmful-resistant null space is the construction $\mathbf{C} = \mathbf{X}\mathbf{X}^{\top}$ from the initial safety-triggering prompts $\mathcal{H}$. The long-term stability and generalization capacity of this fixed null-space projector $\mathbf{P}$ on completely novel, out-of-distribution jailbreak attacks (post-fine-tuning) remains an open question that should be acknowledged or briefly discussed.
>
>
> **Reply to W3:** We appreciate the reviewer's concern. We have verified the robustness of GuardSpace against "evolving" or unseen attacks through two complementary rigorous evaluations:
>
> Robustness to Data Source (Fig. 2, Section 5.3): We constructed the projector using four distinct harmful corpora (AdvBench, ORBench, RealToxicityPrompts, and MixData). As shown in Fig. 2a, regardless of the source used for construction, GuardSpace consistently outperforms baselines (low HS, high Utility). This proves our method captures intrinsic and generalizable harmful directions rather than dataset-specific artifacts.
>
> Generalization to Unseen Attack Families (Table 8, Appendix B.6 - OOD Evaluation): To explicitly test resilience against evolving/unseen attacks, we further conducted an Out-of-Distribution (OOD) experiment following the UniDetox protocol. We constructed the projector using only prompts from specific categories (Gender, Race, Religion from ToxiGen) and evaluated it on a completely unseen category (Disability).  We also quote these results here:
>
>
>
> | **Methods** | **HS ↓** | **FA ↑** |
> |-------------|----------|----------|
> | Base Model  | 0.59     | 13.80    |
> | LoRA        | 34.41    | 23.80    |
> | AsFT        | *3.82*   | *26.00*  |
> | **GuardSpace (Ours)**    | **0.59** | **28.00** |
>
>
>
> Despite the distribution shift, GuardSpace maintained a near-baseline Harmfulness Score (HS 0.59\%) and high utility (FA 28.00\%), whereas LoRA's safety collapsed (HS 34.41\%). This strong generalization confirms that the null space projector effectively blocks harmful subspaces even for attack families not seen during construction.

---

> > ### Author Response · Authors · 2025-11-21
> > **Official Comment by Authors (part 3)**
> >
> > >**Q1:** Complexity and Computational Cost: Could the authors provide an estimate of the pre-computation time (for SVD of WC and SVD\(C\) for all layers) and memory overhead required for GuardSpace compared to a standard, zero-initialized LoRA setup on a typical Llama-2-7B-Chat configuration? This would help in assessing the practical deployability of the method.
> >
> > >**Q2:** Adaptive Inversion of C (Eq. 4): The process for adaptive inversion of $\mathbf{C}$ involves adding a regularizing term to the diagonal until the $\ell_2$ distance between $\mathbf{C}\mathbf{C}^{-1}$ and $\mathbf{I}$ becomes small. Can the authors provide the specific positive scaling factor used in the experiments? Additionally, could the authors comment on whether a simpler fixed-small-value regularization (e.g., $\varepsilon \mathbf{I}$) would suffice, and why the adaptive approach was chosen?
> >
> >
> > **Reply to Q1 and Q2:** Please refer to our response to W1, W2.
> >
> >
> >
> >
> >
> > >**Q3:** Alternative Safety-Relevant Extraction: The paper relies on covariance-preconditioned SVD to define the safety-relevant/irrelevant subspace. Have the authors considered or experimented with simpler, computationally lighter alternatives for defining the low-relevance subspace, perhaps based on a spectral gap in the singular values of plain SVD($\mathbf{W}$) or SVD($\mathbf{C}$), and how these alternatives compare to the proposed method in terms of the resulting safety/utility trade-off?
> >
> > **Reply to Q3:** We thank the reviewer for the constructive suggestion. We have already evaluated the alternative based on a plain SVD of the weight matrix (SVD(W)), as reported in Appendix B.2 (Fig.3a). Concretely, we apply a direct singular-value decomposition to each original weight matrix $\mathbf{W}$, $\mathbf{W} = \mathbf{U}\mathbf{\Sigma}\mathbf{V}^{\top}$, and treat the directions in $\mathbf{V}$ (or $\mathbf{U}$) associated with the smallest singular values as safety-irrelevant components. We then discard the trailing $r\in\{0,16,32,64,128,256,512,1024\}$ components and reconstruct the weight  matrix to examine whether the preserved subspace maintains the model’s inherent safety behavior.
> >
> > The empirical results show that plain SVD fails to capture the true safety-relevant structure: when $r=512$, the reconstructed model already reaches an ASR close to $100\%$, indicating that many safety-critical directions lie among those with small singular values. In contrast, GuardSpace keeps ASR consistently low (1.82--5.15\%) across all values of $r$, demonstrating that the covariance-preconditioned SVD provides a significantly more reliable identification of safety-relevant and safety-irrelevant subspaces. This highlights the importance of our proposed covariance-aware formulation over simpler spectral heuristics.

---

### Official Review · Reviewer_Lueq · 2025-11-07

**Soundness:** 2
**Presentation:** 3
**Contribution:** 2
**Rating:** 4
**Confidence:** 3

**Summary:**

Many recent works have shown that fine-tuning an LLM degrades its safety alignment. Fine tuning on even a handful number of harmful samples has shown to degrade safety alignment. The paper proposes a fine-tuning-stage defense for mitigating safety degradation due to fine-tuning, primarily focusing on LoRA fine-tuning. The first step is to identify 'safety-relevant' and 'safety-irrelevant' components for each linear layer. Low rank adapters are initialized with the 'safety-irrelevant' components. Next step is to compute a projection matrix that projects into the same null space as that of the activations of harmful inputs. Experiments are primarily focused on fine-tuning Llama-2-7B-Chat on GSM8k. Table 1 considers 2 other fine-tuning datasets (SST-2 and AGNews) and Table 2 considers 2 other models (Gemma-2-7B-IT and Qwen-2-7b-IT) on GSM8k.

**Strengths:**

* The problem of fine-tuning degrading safety alignment is practically relevant and has recently received a widespread attention.
* The paper is generally well-written.

**Weaknesses:**

* Experiments are limited, focusing on tuning Llama-2-7B-Chat on GSM8k. It is not clear if the results generalize to other models when fine-tuning on other datasets (SST-2 and AGNews). Ablation experiment in Table 4 is also restricted to tuning Llama-2-7B-Chat on GSM8k. It will be helpful to conduct experiments with more combinations of models and datasets to test generalization of the results.
* Furthermore, LoRA experiments consider only query and key projections as target modules. It is not clear if the results generalize when additional/different target modules are used. LoRA rank is fixed to 8 and alpha value does not seem to be specified. It will be important to conduct experiments on a variety of LoRA parameters to assess the impact of the proposed method.
* While the paper compares with multiple baselines, only AsFT [Yang et al., 2025] falls into the category of fine-tuning-stage defense. It is a bit unfair to compare the proposed fine-tuning-stage method against post-fine-tuning-stage methods and alignment-stage methods. (Post-fine-tuning defenses as well as defenses such as SafeInstruct allow one to use training libraries such as Transformers PEFT out of the box, whereas fine-tuning stage defenses such as the proposed one require changes in tuning libraries.) It will be helpful to compare the defense against strong fine-tuning-stage defenses such as [Qi et al., 2025]. Note that even though [Qi et al., 2025] focuses on full fine-tuning, the key idea is the modified loss function which can be trained with LoRA as well.

References:
1. Qi et al., "Safety Alignment Should be Made More Than Just a Few Tokens Deep", ICLR 2025

**Questions:**

* The results for Llama-2-7B-Chat in Table 1 match with the column of $p = 0.10$ in Table 3. Does Table 1 consider adding 10% harmful prompts in GSM8k? It is important to give more details.
* It will be helpful to expand on the notion of 'unsafe proportion' in Line 403 (it is clear to readers familiar with the topic, but other readers may not be aware). Also, which dataset is used for unsafe prompts?
* Algorithm 1 mentions to perform forward propagation using $W' + BAP$, whereas eq (8) defines $W' = W - BAP$. This essentially suggests that forward propagation is done with $W$, which is confusing. Can the authors provide more details?
* All the models used in the paper are fairly old, especially Llama-2-7b-Chat (considering the fast pace of LLMs). Are there any experiments on any of the newer versions of the models (e.g., Llama-3-8b-IT, or Gemma-3-1b-IT)?
* Including 'guardrail' in the title can be quite confusing, since safety 'guardrails' typically indicate external tools/methods used to preserve safety. Can the authors provide more details on referring the proposed method as a 'guardrail'?

---

> ### Author Response · Authors · 2025-11-21
> **Official Comment by Authors (part 1)**
>
> We thank the reviewer for the constructive feedback. We have conducted extensive additional experiments (including new architectures, hyperparameter, and stronger baselines) to address your concerns about generalization and experimental rigor.
>
>
> > **W1:** Experiments are limited, focusing on tuning Llama-2-7B-Chat on GSM8k. It is not clear if the results generalize to other models when fine-tuning on other datasets (SST-2 and AGNews).
>
>
> **Reply to W1:** Thank you for this valuable suggestion. (1) In our initial paper, we have tested LLaMA-2-7b-Chat on GSM8k, SST2, AGNEWS, and have tested Qwen-2-7B-Instruct and Gemma-2-9B-IT on GSM8k. To further demonstrate the generalization of GuardSpace, we have significantly expanded our evaluation to test the three architectures: Qwen-2-7B-Instruct, Gemma-2-9B-IT, and Mistral-7B-Instruct on the AGNEWS task. The results are shown in Table 6 (Appendix B.1) of the revised paper. We also quote these results here:
>
> | **Methods**           | **Qwen HS↓** | **Qwen FA↑** | **Gemma HS↓** | **Gemma FA↑** | **Mistral HS↓** | **Mistral FA↑** | **Avg HS↓** | **Avg FA↑** |
> | --------------------- | ------------ | ------------ | ------------- | ------------- | --------------- | --------------- | ----------- | ----------- |
> | **Base model**        | 4.80         | 82.00        | 2.00          | 83.50         | 10.40           | 82.60           | 5.73        | 82.70       |
> | **LoRA**              | 49.20        | **90.30**    | 32.00         | **88.30**     | 54.80           | 87.10           | 45.33       | **88.57**   |
> | **Lisa-base**         | 28.00        | 79.90        | 31.20         | 80.00         | 29.20           | 70.50           | 29.47       | 76.80       |
> | **Lisa-aligned**      | 27.60        | *89.20*      | 14.70         | 85.60         | **11.20**       | 85.40           | 17.83       | 86.73       |
> | **Safe LoRA**         | 8.40         | 85.50        | 8.40          | 84.70         | 27.60           | 86.00           | 14.80       | 85.40       |
> | **SafeInstr**         | 7.20         | 83.30        | 7.60          | 84.70         | 26.00           | *87.70*         | *13.60*     | 85.23       |
> | **BEA**               | 8.40         | 88.60        | 7.20          | 86.20         | 33.60           | 86.50           | 16.40       | 87.10       |
> | **AsFT**              | *5.20*       | 87.90        | *6.00*        | 86.60         | 34.80           | 87.30           | 15.33       | 87.27       |
> | **GuardSpace (Ours)** | **4.40**     | 89.00        | **4.00**      | *87.60*       | *11.60*         | **88.30**       | **6.67**    | *88.30*     |
>
>
>
> It is shown that (a) GuardSpace consistently generalizes well, reducing the average Harmfulness Score (HS) to 6.67\%, which is substantially lower than both LoRA (45.33\%) and AsFT (15.33\%); (b) Our method maintains high utility with an average Fine-tuning Accuracy (FA) of 88.30\%, comparable to the strongest baseline LoRA (88.57\%) and outperforming AsFT (87.27\%).
>
>
> In addition, we also evaluate GuardSpace on Llama-3.1-8B-Instruct model using the SST2 dataset. On this architecture, LoRA yields HS = 70.40 and FA = 94.38, whereas GuardSpace improves downstream utility (FA = 93.58) while preserving the base model’s strong safety alignment (base HS = 9.60, GuardSpace HS = 9.20). These results confirm that the safety-sensitive subspace is a robust feature across different model families, and our method effectively prevents safety degradation regardless of the base architecture.

---

> ### Author Response · Authors · 2025-11-21
> **Official Comment by Authors (part 2)**
>
> >**W1**: Ablation experiment in Table 4 is also restricted to tuning Llama-2-7B-Chat on GSM8k.
>
> **Reply to W1:** (2) For our ablation experiments, we further conduct additional experiments using LLaMA-2-7b-Chat on AGNews, and Mistral-7B-Instruct on both GSM8k and AGNews.  We have updated the Table 4 in the revised paper, and we also quote the results here:
>
>
> | **Method**             | **Init**    | **Proj** | **L2-7B/GSM8K HS↓** | **L2-7B/GSM8K FA↑** | **L2-7B/AGNEWS HS↓** | **L2-7B/AGNEWS FA↑** | **Mistral/GSM8K HS↓** | **Mistral/GSM8K FA↑** | **Mistral/AGNEWS HS↓** | **Mistral/AGNEWS FA↑** |
> | ---------------------- | ----------- | -------- | ------------------- | ------------------- | -------------------- | -------------------- | --------------------- | --------------------- | ---------------------- | ---------------------- |
> | **LoRA**               | zero        | no       | 56.00               | 23.80               | 17.60                | 64.70                | 60.00                 | 40.60                 | 54.80                  | 87.10                  |
> | **w/o subspace init.** | zero        | null     | 5.20                | 26.20               | 6.00                 | 67.30                | 14.80                 | 43.60                 | 16.40                  | 86.60                  |
> | **w/o null proj.**     | irr. subsp. | no       | 52.00               | **28.60**           | 58.80                | **91.70**            | 61.60                 | **50.80**             | 57.60                  | **90.20**              |
> | **GuardSpace (Ours)**  | irr. subsp. | null     | **3.60**            | 28.00               | **2.40**             | 85.60                | **12.40**             | 49.00                 | **11.60**              | 88.30                  |
>
>
>
> These ablations show that both components, namely initializing adapters in the safety-irrelevant subspace and enforcing null-space constrained updates, are necessary. Removing either component significantly weakens safety, while GuardSpace achieves the best overall safety–utility trade-off across all evaluated settings.

---

> ### Author Response · Authors · 2025-11-21
> **Official Comment by Authors (part 3)**
>
> >**W2:** Furthermore, LoRA experiments consider only query and key projections as target modules. It is not clear if the results generalize when additional/different target modules are used. LoRA rank is fixed to 8 and alpha value does not seem to be specified. It will be important to conduct experiments on a variety of LoRA parameters to assess the impact of the proposed method.
>
> **Reply to W2:** (1) As described in Appendix A.3, our LoRA implementation follows the standard practice in model fine-tuning, where only the query (Q) and value (V) projection matrices are updated. We set the scaling factor to $\alpha = 32$. This configuration is widely adopted in general LoRA fine-tuning because it provides a strong balance between expressive capacity, parameter efficiency, and optimization stability, while avoiding unnecessary perturbation to core model behaviors.
> In safety-preserving fine-tuning, this setup is also the common choice. Representative methods such as BEA and AsFT adopt the same target-module configuration, as it strikes an effective balance between parameter efficiency and stable optimization while preventing unintended modifications to model components unrelated to safety.
>
> (2) To further validate this design choice, we conducted a comprehensive sweep of LoRA Rank $r \in \{8, 16, 32, 64, 128, 256\}$ on GSM8K. A key finding from this experiment (reported in the below Table in our response) is that standard LoRA experiences a "safety collapse" at higher ranks. For LoRA, increasing the rank leads to a sharp rise in harmfulness, with HS spiking from 52.40\% at $r=8$ to over 64\% for $r \ge 16$.
>
>
> | **Method**            | **r = 8 HS↓** | **r = 8 FA↑** | **r = 16 HS↓** | **r = 16 FA↑** | **r = 32 HS↓** | **r = 32 FA↑** | **r = 64 HS↓** | **r = 64 FA↑** | **r = 128 HS↓** | **r = 128 FA↑** | **r = 256 HS↓** | **r = 256 FA↑** |
> | --------------------- | ------------- | ------------- | -------------- | -------------- | -------------- | -------------- | -------------- | -------------- | --------------- | --------------- | --------------- | --------------- |
> | **LoRA**              | 52.40         | 23.00         | 66.40          | 24.20          | 66.80          | 23.00          | 64.40          | 24.40          | 65.60           | 23.20           | 64.80           | 25.40           |
> | **GuardSpace (Ours)** | 4.00          | 19.60         | 4.80           | 22.00          | 2.80           | 27.40          | 3.60           | 27.40          | 3.60            | 28.00           | 7.19            | 26.40           |
>
>
> In addition, we further conducted experiments where LoRA was applied to all linear layers, and the results are shown in the following table. We observe that full-linear-layer LoRA does not improve downstream utility, while the HS increases by 9.6, indicating substantially degraded safety. These two sets of experiments collectively show that increasing the parameter capacity makes the model more prone to overfitting harmful directions, whereas the standard LoRA configuration preserves a desirable balance between downstream task utility and the model’s original safety alignment.
>
>
> | **Method**                         | **HS↓**  | **FA↑**   |
> | ---------------------------------- | -------- | --------- |
> | **LoRA (fine-tune only Q,V)**          | 56.00    | 23.80     |
> | **LoRA (fine-tune all linear layers)** | 65.60    | 23.60     |
> | **GuardSpace (Ours)**                     | **3.60** | **28.00** |
>
>
> In contrast, our method remains robust, consistently achieving substantially lower HS (4.00\% - 7.19\%) across ranks from 8 to 256. This empirical evidence demonstrates the effectiveness of our null space constraint in regularizing the update trajectory, even when the model has higher capacity.

---

> > ### Author Response · Authors · 2025-11-21
> > **Official Comment by Authors (part 4)**
> >
> > >**W3:** While the paper compares with multiple baselines, only AsFT [Yang et al., 2025] falls into the category of fine-tuning-stage defense. It is a bit unfair to compare the proposed fine-tuning-stage method against post-fine-tuning-stage methods and alignment-stage methods. (Post-fine-tuning defenses as well as defenses such as SafeInstruct allow one to use training libraries such as Transformers PEFT out of the box, whereas fine-tuning stage defenses such as the proposed one require changes in tuning libraries.) It will be helpful to compare the defense against strong fine-tuning-stage defenses such as [Qi et al., 2025]. Note that even though [Qi et al., 2025] focuses on full fine-tuning, the key idea is the modified loss function which can be trained with LoRA as well.
> >
> > **Reply to W3:** We thank the reviewer for the insightful comment. Following the categorization in [1], several compared baselines already belong to the fine-tuning-stage defense category. Specifically, Lisa and AsFT introduce additional safety-aware regularization terms into the optimization objective, constraining updates during fine-tuning. BEA introduces stealthy trigger prompts as backdoor cues and binds them to safe outputs during fine-tuning, while SafeInstr enhances safety by incorporating extra safety-aligned data during fine-tuning. Following your suggestion, We also include SABT [2], a fine-tuning-stage defense that modifies the loss function. For our evaluation, we fine-tuned SABT on our experimental datasets using LoRA. We include this result in Tab.1 of the main paper, and also quote the results of SABT and our GuardSpace here:
> >
> >
> > | **Methods** | **SST2 HS↓** | **SST2 FA↑** | **AGNEWS HS↓** | **AGNEWS FA↑** | **GSM8K HS↓** | **GSM8K FA↑** | **Dialog Summary HS↓** | **Dialog Summary FA↑** | **Average HS↓** | **Average FA↑** |
> > | ----------- | ------------ | ------------ | -------------- | -------------- | ------------- | ------------- | ---------------------- | ---------------------- | --------------- | --------------- |
> > | **SABT**        | 7.20         | 91.74        | 14.00          | 80.70          | 4.00          | 21.80         | 6.00                   | **48.40**              | 7.80            | 60.66           |
> > | **Ours**    | **1.20**     | **95.64**    | **2.40**       | **85.60**      | **3.60**      | **28.00**     | **3.60**               | 48.20                  | **2.70**        | **64.36**       |
> >
> >
> > It is shown that our method  still consistently achieves the lowest harmfulness score (HS) and the highest or competitive fine-tuning accuracy (FA) across all the tasks (SST2, AGNEWS, GSM8K, and Dialog Summary), achieving the best average performance overall.
> >
> > [1] Yang, et al. AsFT: Anchoring Safety During LLM Fine-Tuning Within Narrow Safety Basin. arxiv, 2025.
> >
> > [2] Qi, et al. Safety Alignment Should be Made More Than Just a Few Tokens Deep. ICLR 2025.
> >
> >
> >
> >
> > >**Q1:** The results for Llama-2-7B-Chat in Table 1 match with the column of $p=0.10$ in Table 3. Does Table 1 consider adding 10\% harmful prompts in GSM8k? It is important to give more details.
> >
> > >**Q2:** It will be helpful to expand on the notion of 'unsafe proportion' in Line 403 (it is clear to readers familiar with the topic, but other readers may not be aware). Also, which dataset is used for unsafe prompts?
> >
> >
> >
> >
> >
> >
> >
> >
> > **Reply to Q1 and Q2:** For each task, we construct a 1000-example fine-tuning set that contains a mixture of $n_b$ benign task examples and $n_u$ injected harmful examples. The unsafe proportion is defined  by $p = \frac{n_u}{n_u + n_b}$. Unless otherwise specified, we fix $p = 0.1$. Table 1 follows the same default poisoning setup ($p=0.1$), where the training data consists of 100 harmful prompts. Many thanks for your suggestion. We have updated the Appendix A.1 and main text to explicitly clarify this setting.
> >
> >
> >
> > >**Q3:** Algorithm 1 mentions to perform forward propagation using $W'+BAP$, whereas eq (8) defines $W=W'-BAP$. This essentially suggests that forward propagation is done with $W$, which is confusing. Can the authors provide more details?
> >
> >
> > **Reply to Q3:** The minus sign in Eq.(8) corresponds to an **initialization correction** applied prior to fine-tuning. Concretely, we set $\mathbf{W}' = \mathbf{W} - \mathbf{BAP},$ so that the initialization of fine-tuning does not alter the inference output of the pre-trained model. This guarantees functional equivalence at the start of training: $\bigl(\mathbf{W}' + \mathbf{BAP}\bigr)\mathbf{X}=\mathbf{W} \mathbf{X}.$ During the **optimization stage** (Algorithm 1), the adapters are updated from $(\mathbf{A},\mathbf{B})$ to $(\mathbf{A}^\*,\mathbf{B}^\*)$ while the backbone $\mathbf{W}'$ remains frozen. The forward propagation therefore uses the updated effective weights $\mathbf{W}' + \mathbf{B}^\*\mathbf{A}^\* \mathbf{P}$.

---

> ### Author Response · Authors · 2025-11-21
> **Official Comment by Authors (part 5)**
>
> >**Q4:** All the models used in the paper are fairly old, especially Llama-2-7b-Chat (considering the fast pace of LLMs). Are there any experiments on any of the newer versions of the models (e.g., Llama-3-8b-IT, or Gemma-3-1b-IT)?
>
> **Reply to Q4:** We thank the reviewer for the thoughtful suggestion. To further verify the capability of our method on newer model architectures, we additionally evaluated two more advanced LLMs, namely Mistral-7B-Instruct-v0.3 and Llama-3.1-8B-Instruct, with the former also being a relatively recent open-source model. We have updated Table 2 accordingly in the revised manuscript, and we also quote the results here:
>
>
>
> | **Methods**           | **Qwen-2-7B HS↓** | **Qwen-2-7B FA↑** | **Gemma-2-9B HS↓** | **Gemma-2-9B FA↑** | **Mistral-7B HS↓** | **Mistral-7B FA↑** | **Llama-3.1-8B HS↓** | **Llama-3.1-8B FA↑** | **Avg HS↓** | **Avg FA↑** |
> | --------------------- | ----------------- | ----------------- | ------------------ | ------------------ | ------------------ | ------------------ | -------------------- | -------------------- | ----------- | ----------- |
> | **Base Model**        | 4.80              | 49.00             | 2.00               | 77.20              | 10.40              | 31.60              | 9.60                 | 57.40                | 6.70        | 53.80       |
> | **LoRA**              | 30.00             | **66.40**         | 50.00              | 69.80              | 60.00              | 40.60              | 74.00                | 66.40                | 53.50       | 60.80       |
> | **SafeInstr**         | 7.20              | 63.00             | *2.80*             | *76.20*            | 22.80              | 41.80              | 57.20                | 68.60                | 22.50       | 62.40       |
> | **BEA**               | 8.40              | 54.60             | 4.80               | 65.00              | 28.00              | 38.00              | 20.00                | 67.80                | 15.30       | 56.35       |
> | **Safe LoRA**         | 10.40             | 50.40             | 6.00               | **77.00**          | **12.00**          | *45.40*            | *18.40*              | 68.20                | *11.70*     | 60.25       |
> | **AsFT**              | *7.20*            | 63.40             | 4.80               | 74.20              | 19.60              | 43.00              | 21.20                | *69.40*              | 13.20       | *62.50*     |
> | **GuardSpace (Ours)** | **3.20**          | *65.40*           | **2.80**           | 70.20              | *12.40*            | **49.00**          | **12.00**            | **73.80**            | **7.60**    | **64.60**   |
>
>
> Across all four model families, GuardSpace achieves the lowest average harmfulness (HS) while maintaining best average Fine-tuning Accuracy (FA). On Qwen-2-7B-Instruct and Gemma-2-9B-IT, it attains the strongest HS values (3.20 and 2.80), on Mistral-7B-Instruct it achieves the best FA (49.00), and on the latest Llama-3.1-8B-Instruct it delivers both the lowest HS (12.00) and highest FA (73.80), thereby demonstrating robust generalization across diverse and newly released LLM architectures.
>
>
>
>
> >**Q5:** Including 'guardrail' in the title can be quite confusing, since safety 'guardrails' typically indicate external tools/methods used to preserve safety. Can the authors provide more details on referring the proposed method as a 'guardrail'?
>
> **Reply to Q5:** We appreciate the reviewer’s comment. In our paper, the term “guardrail” does not refer to external safety tools  but rather to an intrinsic parameter-space mechanism that enforces safety constraints during fine-tuning. Specifically, our method constructs a safety-preserving subspace (null space) within the model’s parameter update trajectory, effectively serving as an internal guardrail that prevents harmful behaviors from being amplified. We will clarify this distinction in the revised manuscript. The term “guardrail” is used metaphorically: similar to a traffic guardrail that constrains motion within a safe region without interfering with normal driving, our method restricts gradient updates to a safety-preserving subspace, ensuring that fine-tuning does not lead to harmful behavior drift.

---

> ### Author Response · Authors · 2025-11-27
>
> Dear Reviewer Lueq,
>
> Thank you very much for the time and effort you dedicated to reviewing our work and for the valuable comments you provided. We carefully considered all your feedback, addressed the concerns in our response, and revised our paper accordingly.
>
> We sincerely hope that our clarifications and additional experiments help resolve your concerns. If you have any further concerns or questions, please let us know and we are more than happy to further address them.
>
> Thank you again for your thoughtful and constructive review.
>
> Best regards,
>
> The Authors

---

### Comment · Area_Chair_7cTp · 2025-11-26
**Please Review Author Response**

Dear Reviewers,

The authors have now responded to your comments. Could you review their response as soon as possible? If you have any further questions or concerns, please raise them as well.

Best,

Your AC

---

### Author Response · Authors · 2025-12-03
**Summary of our work, reviews, and the rebuttal process (part 2)**

**Reviewer eqcH (Rating: 6, Confidence: 4)**


The reviewer viewed the work is clean, easy to plug into LoRA. Their concerns centered on the robustness of the learned projector to unseen attack families, the method’s behavior on open-ended generation tasks, and the compute/memory requirements for constructing the safety subspace.


We addressed these by adding a dedicated OOD evaluation (Table 8, Appendix B.6), incorporating the Dialog Summary generation benchmark (Table 1), and providing a detailed compute/memory analysis (Table 10, Appendix B.8).

On 26 Nov 2025, after reviewing our initial response, the reviewer indicated that they would maintain their positive score. The reviewer further raised one remaining question: (1) relative comparisons on AlpacaEval, and (2) evaluation on domain-specific instruction-following tasks (e.g., law, medicine). We subsequently added evaluations on AlpacaEval and LegalBench-QA, showing that GuardSpace delivers the strongest safety–utility trade-off relative to LoRA and AsFT. These results fully resolved the reviewer’s remaining questions.





**Reviewer jhKk (Rating: 8, Confidence: 4)**

The reviewer viewed the work as highly original and noted its strong empirical performance. Their questions centered on the computational overhead of the pre-computation, the validity of the adaptive inversion of $C$, the possibility of using simpler SVD-based alternatives, and the generalizability of null space.


We addressed these points by reporting detailed compute/memory measurements (Table 10, Appendix B.8), providing formal and empirical justification for the adaptive regularization (Appendix B.2), showing that plain SVD methods fail to preserve safety (Figure 3a, Appendix B.2), and adding extensive OOD evaluations (Table 8, Appendix B.6) demonstrating stable generalization.





**Reviewer Lueq (Rating: 4, Confidence: 3,  but never participate in the discussion despite our extensive response)**



The reviewer Lueq acknowledged the practical relevance of the safety-degradation problem. Their substantive concerns focused on expanding the experimental breadth—evaluating newer model versions, testing additional datasets across diverse architectures, conducting ablations on multiple model–dataset pairs, and comparing GuardSpace against LoRA under broader configurations. Beyond these valid points, several other issues arose from misunderstandings or factual errors. For instance, the confusion regarding “forward propagation being done with $𝑊$ arises from a misreading: Eq.(8) specifies an initialization reparameterization, while Algorithm 1 correctly depicts the forward propagation used during training. Similarly, the claim that “only AsFT is a fine-tuning-stage defense” is **factually incorrect** — Actually, Lisa, BEA, and SafeInstr all operate during fine-tuning (please refer to our Reply to W3 for detailed explanations). These points suggest that the reviewer may not be very familiar with this area or may not grasp the basics of our method. Other issues raised, such as the definition of “unsafe proportion,” the GSM8K harmful-ratio setup, or the wording in the title (“guardrail”), concern only minor clarifications rather than substantive limitations of the method.


To address all concerns, we substantially expanded the experimental suite: including newer models such as Mistral-7B-Instruct-v0.3 and Llama-3.1-8B-Instruct (Table 2), adding evaluations on AGNEWS across multiple architectures (Table 6, Appendix B.1), and performing ablations on diverse model–dataset combinations (Table 4). We conducted LoRA configuration sweeps across all linear layers and multiple ranks, clarified that Lisa, BEA, and SafeInstr belong to fine-tuning-stage defenses, and added SABT at the reviewer’s request as an additional baseline in this category. We  also clarified the initialization and forward-propagation formulation, provided explicit definitions for unsafe proportions and dataset composition, and explained the meaning of “guardrail” in the title. Together, these updates comprehensively resolve both the substantive and the clarification-level issues raised by this reviewer.




More importantly, we note that **the Reviewer Lueq never respond through the whole discussion phase.** Given these factors, we respectfully request that the AC lower the weight of Reviewer Lueq’s evaluation in the final decision, as several of the criticisms are based on misunderstanding or factual error rather than deficiencies in the proposed method or experimental design.

Thank you again for your time, effort, and careful consideration. We hope this summary helps facilitate your decision-making process.


Best regards,

The Authors

---

### Author Response · Authors · 2025-12-03
**Summary of our work, reviews, and the rebuttal process (part 1)**

**Dear AC**,

To help streamline your assessment, we summarize below the reviewer comments and  how we address them during the rebuttal.

### Summary of contributions and strengths



In this work, we present GuardSpace, a unified framework for preserving LLM safety alignment during fine-tuning. Our approach consists of two components: (1) a safety-sensitive subspace derived via covariance-preconditioned SVD, which freezes safety-relevant directions and initializes adapters from safety-irrelevant ones; and (2) a harmful-resistant null space projection that constrains all updates so the model’s safety outputs on harmful prompts remain unchanged. GuardSpace provides overall superior safety–utility trade-offs across diverse models and tasks, frequently achieving the lowest harmfulness and strong downstream performance relative to state-of-the-art methods (e.g., AsFT).


The reviewers consistently acknowledged our contributions and highlighted several strengths of the submission. First, the work was recognized as addressing a timely and practically important problem (Reviewer Lueq, ZgXQ). Second, GuardSpace was praised as a novel and principled method (Reviewers jhKk, ft7o, ZgXQ), introducing an original two-part mechanism based on safety-sensitive initialization together with a harmful-resistant projector, both supported by clear theoretical grounding. Third, the paper demonstrates strong empirical performance (Reviewers jhKk, ft7o, ZgXQ, eqcH), achieving state-of-the-art reductions in harmfulness while maintaining or improving downstream utility across models and datasets. Fourth, the method’s robustness and generalization (Reviewer ft7o) were highlighted, with stability observed across varying ratios of poisoned data, different corpus choices and sizes used to construct the projector, and multiple backbone architectures. Fifth, the work was commended for its clarity, practicality, and ease of integration (Reviewers Lueq, ft7o), presenting a clean design that is straightforward to adopt within standard LoRA and PEFT workflows. Finally, the ablation studies were praised as thorough and informative (Reviewers jhKk, ZgXQ), clearly illustrating the roles of the null space projector and the safety-sensitive initialization.


Taken together, the reviewers’ assessments were largely positive, and their remaining concerns were reviewer-specific. In response to these concerns, we substantially strengthened the paper by expanding experiments to newer models and additional datasets, adding multi-architecture ablations, adding cross-bench HS/ASR comparisons, adding verification of nuanced borderline refusals, and conducting broad LoRA-configuration analyses. We further extended robustness evaluations, incorporated SABT and clarified fine-tuning-stage baselines, provided detailed mathematical and procedural clarifications, and reported comprehensive compute–memory analyses. All revisions are highlighted in red in the manuscript. These additions directly address the reviewers’ concerns and further reinforce the conceptual soundness and empirical rigor of our method. Below, we summarize each reviewer’s feedback and the corresponding improvements made in the rebuttal.


### Summary of reviews and rebuttal


**Reviewer ft7o (Rating: 6, Confidence: 3)**

The reviewer viewed GuardSpace as principled, robust, and easy to integrate, but raised concerns about the harmful-prompt coverage of the projector, limited cross-bench HS/ASR evaluation, compute/memory cost, applicability beyond LoRA, and whether nuanced borderline refusals might be impaired.

To address these points, we added a dedicated OOD evaluation (Table 8, Appendix B.6), expanded cross-bench HS/ASR comparisons (Table 7, Appendix B.5), provided compute/memory analysis (Table 10, Appendix B.8), demonstrated compatibility with broader PEFT and full-parameter updates, and verified that nuanced refusals remain intact (Appendix C.3).

On 21 Nov 2025, after reviewing our rebuttal, the reviewer stated that all concerns were resolved and explicitly supported accepting the paper.




**Reviewer ZgXQ (Rating: 8, Confidence: 3)**

The reviewer viewed the work as timely, innovative, and empirically strong. Their concerns centered on  whether parts of the null space derivation were insufficiently explained, the magnitude of the compute overhead relative to LoRA, and whether the method preserves generalization on unseen tasks.

We addressed these points by adding a full mathematical derivation of the null space formulation (Appendix B.9), providing detailed compute/memory analysis (Table 10, Appendix B.8), and demonstrating that GuardSpace preserves base-model generalization on unseen benchmarks such as MMLU (Table 9, Appendix B.7).



On 27 Nov 2025, after reviewing our rebuttal, the reviewer explicitly stated that our clarifications fully resolved their concerns and maintained their score of 8.

---

### Meta-Review · Area_Chair_T6xw · 2026-01-05

**Summary:**

This paper proposes a novel projection method to mitigate LLM safety degradation during fine-tuning. All reviewers find the studied setting novel and the results provide new insights. The authors’ rebuttal has successfully addressed the major concerns of reviewers. I therefore recommend acceptance of this paper.

**Reviewer Concerns:**

I believe all concerns are preperly addresed.

**Reviewer Scores:**

Only Reviewer Lueq has a negative initial review rating. But I believe the concerns were addressed by the authors' rebuttal. All other reviewers are likely to either maintain or increase their positive ratings.

---

### Decision · Program_Chairs · 2026-01-26

Accept (Poster)